# Free charge photogeneration in a single component high photovoltaic efficiency organic semiconductor

Michael B. Price [1,2,10 ✉], Paul A. Hume [1,2,10 ✉], Aleksandra Ilina [1,2], Isabella Wagner [1,2], Ronnie R. Tamming[1,2,3,4,5], Karen E. Thorn[1,2], Wanting Jiao [6], Alison Goldingay [7], Patrick J. Conaghan[7], Girish Lakhwani [7], Nathaniel J. L. K. Davis[1,2], Yifan Wang[8,9], Peiyao Xue[8], Heng Lu[8], Kai Chen[1,2,3,4,5], Xiaowei Zhan[8] & Justin M. Hodgkiss [1,2 ✉]

Organic photovoltaics (OPVs) promise cheap and flexible solar energy. Whereas light generates free charges in silicon photovoltaics, excitons are normally formed in organic semiconductors due to their low dielectric constants, and require molecular heterojunctions to split into charges. Recent record efficiency OPVs utilise the small molecule, Y6, and its analogues, which – unlike previous organic semiconductors – have low band-gaps and high dielectric constants. We show that, in Y6 films, these factors lead to intrinsic free charge generation without a heterojunction. Intensity-dependent spectroscopy reveals that 60–90% of excitons form free charges at AM1.5 light intensity. Bimolecular recombination, and hole traps constrain single component Y6 photovoltaics to low efficiencies, but recombination is reduced by small quantities of donor. Quantum-chemical calculations reveal strong coupling between exciton and CT states, and an intermolecular polarisation pattern that drives exciton dissociation. Our results challenge how current OPVs operate, and renew the possibility of efficient single-component OPVs.

[1] School of Chemical and Physical Sciences, Victoria University of Wellington, Wellington, New Zealand. [2] MacDiarmid Institute for Advanced Materials and Nanotechnology, Wellington, New Zealand. [3] Wellington UniVentures, Victoria University of Wellington, Wellington, New Zealand. [4] Robinson Research Institute, Faculty of Engineering, Victoria University of Wellington, Wellington, New Zealand. [5] The Dodd-Walls Centre for Photonic and Quantum Technologies, Dunedin, New Zealand. [6] Ferrier Research Institute, Victoria University of Wellington, Wellington, New Zealand. [7] ARC Centre of Excellence in Exciton Science, School of Chemistry, University of Sydney, Sydney, NSW, Australia. [8] School of Materials Science and Engineering, Peking University, Beijing, China. [9] College of Materials Science and Engineering, Qingdao University, Qingdao, China. [10] These authors contributed equally: Michael B. Price, Paul A. Hume. ✉email: michael.price@vuw.ac.nz; paul.hume@vuw.ac.nz; justin.hodgkiss@vuw.ac.nz

When light is absorbed by inorganic semiconductors, like silicon or gallium arsenide, a combination of free charges and excitons (bound electrons and holes) are created, and their ratio depends upon the material's dielectric constant. For organic semiconductors, the paradigm holds that only excitons are intrinsically photogenerated because the low dielectric constant[1,2] of these materials ($\varepsilon \sim$ 3–4) means the electron and hole Coulombic interaction is not efficiently screened, leading to Frenkel excitons with high binding energies ($E_B$). For the last thirty years[3,4], this constraint has guided the development of organic photovoltaics (OPVs) - which promise a step-change in flexible, lightweight, non-toxic solution-processed solar energy production, but are yet to be widely commercialised. Splitting bound excitons into free charges has required sharp molecular heterojunctions between donor and acceptor materials. Rather than purely optimising charge harvesting, OPV devices have been optimised for exciton splitting at interfaces, and demand complex interpenetrating networks of donor and acceptor materials. The molecular heterojunction approach limits device efficiencies[5], introduces inherent voltage losses and instabilities from interfaces[6], and complicates research progress[7]. Reports have previously shown some field-assisted exciton dissociation, and 'extrinsic' charge formation in neat homopolymers and small molecules[8–11], as well as significant charge-transfer (or polaron-pair) state formation in neat "push-pull" copolymers and small molecules[12–14]. However no organic material (until now) has exhibited substantial, let alone majority, free charge (rather than polaron pair) formation – which undergoes bimolecular (rather than geminate), recombination.

Recently, small molecule non-fullerene fused ring electron acceptors (FREAs)[15–21] have driven a rapid uptick in the power conversion efficiency (PCE) of OPVs. Alongside this advance has arisen unexpected observations, namely: (1) Barrierless free charge generation in PM6:Y6 blends[22] (full chemical names and structures shown in Supplementary Fig. 1). (2) Charge generation efficiency in blends of PM6:Y6 increases with increasing incident light intensity[23]. (3) In Y6[11,24] and IDIC[25], excitons are delocalised, or form an 'intra-moiety' intermediate state with likely charge-transfer (CT) like character. These observations have all been explained by invoking CT-states, however, with additional data, we show here that a more profound explanation is required.

FREAs have recently been measured to have very high refractive indices[26,27], and hence high complex dielectric constants. Of the FREAs, Y6 and its derivatives are present in both the binary and ternary OPVs that hold the highest PCEs[16,28–30].

In inorganic semiconductors, the number of bound excitons versus free charges can be described by the Saha-Langmuir relation.

$$\frac{x^2}{1-x} = \frac{1}{n}\left(\frac{2\pi m^* k_B T}{h^2}\right)^{1.5} e^{-\frac{E_B}{k_B T}} \tag{1}$$

where the free charge fraction, $x$, is dependent upon the temperature, $T$, excitation density, $n$, effective mass, $m^*$ and the exciton binding energy, $E_B$ (which is related to the inverse of the dielectric constant[31]). From comparison of the highest occupied molecular orbital (HOMO)-lowest unoccupied molecular orbital (LUMO) gap to the optical gap measured by Karuthedath et al.[6], the exciton binding energy of Y6 is $E_B$ = 100–250 meV - much lower than, for example, P3HT ($E_B \sim$ 700 meV)[32]. Based on a midpoint, 175 meV binding energy, estimated for Y6 above, and assuming an effective mass between[33,34] $m^*$ = 1.7–0.2 $m_e$, this simple calculation would predict a high fraction (20–80%) of intrinsically generated free charges.

Here, we show that the high optical frequency dielectric constant of Y6 leads to majority free charge generation upon optical excitation within neat films. To prove that the aforementioned

'intra-moiety' TA signature is in fact due to free charges, we demonstrate that recombination follows bimolecular kinetics, rather than monomolecular decay as would occur for CT states. We use a combination of ultrafast transient absorption, photoluminescence up-conversion and time-resolved terahertz spectroscopy, and identify the spectral signatures and decay rates of singlet excitons and free charges (polarons). Intensity-dependent photoluminescence (PL) measurements prove the existence of radiative bimolecular charge recombination, which would not be the case if only CT states were formed. We present simple kinetic models that accurately reproduce our experimental data, wherein a free charge fraction of between 0.7–0.99 exists in equilibrium with bound charges under steady-state illumination, and 60–90% of photoexcited excitons dissociate to charges. Bimolecular, and rapid minority carrier, charge recombination inhibit the efficiency of single-component devices, but this recombination can be slowed by 'doping' our Y6 films with only a small amount of 'p-type' polymer electron donor. Using quantum-chemical calculations, we show that free charge generation is expected due to strong coupling between equienergetic exciton and CT states, and reveal that asymmetric polarisation of the different lattice positions results in a "donor/acceptor" system derived purely from intermolecular interactions.

## Results

**Radiative bimolecular recombination.** Fig. 1a shows the normalised excitation density-dependent external photoluminescence quantum efficiency (PLQE) of two neat films of Y6 (BTP-4F) of differing thicknesses. As the intensity of pulsed excitation is increased, the PLQE increases steadily, peaks, and then decreases for both films. This peak in PL efficiency has not been observed before in similar organic systems, where it is expected that the PL remains steady with increasing fluence before dropping in efficiency as non-radiative bimolecular processes kick-in. Three explanations for the phenomena of a rise in PL with increasing pump intensity are possible: saturation of exciton (or charge) traps, triplet-triplet annihilation (TTA), or bimolecular radiative recombination of intrinsically generated free charges (bulk ionisation).

Exciton trap saturation is considered in detail in the Supplementary Information. We rule out this explanation due to the unphysical rate constants required for it to match the transient absorption and PL data, and due to the lack of any other evidence of exciton trap saturation in the TA spectra/kinetics. TTA is also considered in the Supplementary Information. Due to a number of considerations, such as the large numbers of triplets required for this effect to be solely responsible for the PL rise, which conflicts with experimental evidence, we also rule this out. We therefore conclude that a substantial fraction of photogenerated free charges must be responsible for the PL rise shown in Fig. 1a. As detailed in the below sections, and the Supplementary Information, charge trap saturation, and TTA are possible, and likely present to some extent in our system, but only in conjunction with significant intrinsic charge generation.

Figure 1b illustrates the expected behaviour of excitation density-dependent PLQE for varying proportions of radiative monomolecular to bimolecular processes. The rise in efficiency observed in Fig. 1a is not possible for a system undergoing monomolecular (geminate) radiative recombination, even from intra-moiety CT states, and exciton-exciton annihilation can only produce a bimolecular contribution with reduced radiative efficiency. Charges must be present in the neat film, and these charges are formed 'intrinsically', rather than through exciton annihilation processes. Under steady-state illumination, this free charge (polaron) population will exist concurrently with a

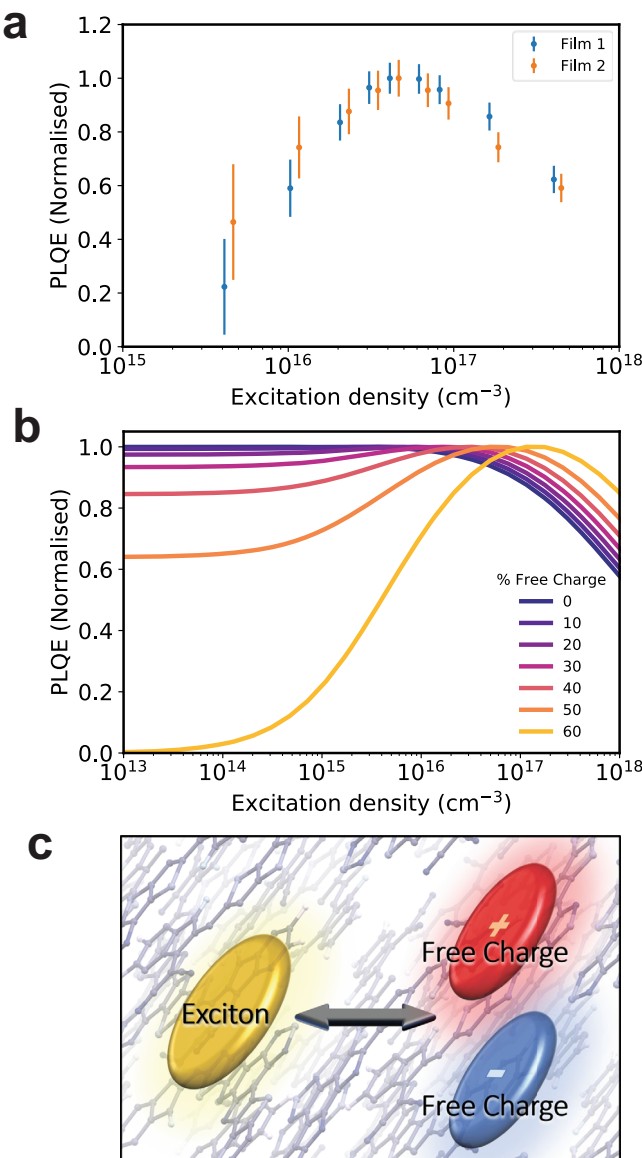

**Fig. 1 Intensity-dependent photoluminescence quantum efficiency of Y6.**
**a** Normalised photoluminescence efficiency, as a function of excitation density, of 130 nm thick (blue dots) and 80 nm thick (orange dots) neat films of Y6 on a glass substrate. The sample was excited by 600 ps pulses of 532 nm light. The sample was measured in vacuum, and the relative photoluminescence efficiency was determined from the ratio of the 2nd harmonic scatter peak at 1064 nm to the integrated PL intensity between 850 and 1050 nm. The error bars shown represent the relative error of the change in PL at different excitation densities, rather than the total error (details in the Supplementary Information). **b** Simulation of intensity-dependent PLQE values showing effect of increasing (orange) or decreasing (blue) radiative bimolecular recombination (from prompt free charge fraction, calculated using the model detailed in the text and SI. **c** Graphic representation of exciton-free charge equilibrium in Y6.

population of bound Frenkel or CT-like excitons, as illustrated by Fig. 1c.

**Exciton - charge dynamics**. To quantify the relative proportions of charges versus excitons, we perform ultrafast spectroscopic measurements on films of neat Y6. We identify spectral signatures of the singlet exciton, and free charge/CT-state species

and their kinetics to estimate first and second order kinetic rates/lifetimes. Figure 2a shows the area-normalised transient absorption (TA) spectrum of a neat Y6 film pumped with a 150 fs, 700 nm laser pulse at a moderate excitation density of ~$10^{17}$ excitations/cm³. The spectrum evolves from one initial photo-excited species, a sharp negative peak at 915 nm, to another consisting of both positive and negative features, apparent after ~500 ps. Singular value decomposition, multivariate curve resolution analysis and application of a genetic algorithm (GA)[35] consistently resolve two major species present, shown as the blue and orange spectra in Fig. 2b. We confirm that the orange species is that of the polaron – referred to as free charge (FC) for consistency in the rest of the manuscript – by comparing the TA spectra of Y6 blended with two hole accepting species (PTB7-Th and poly-TPD), to reveal the Y6 electron signature. The charge spectrum in neat Y6 closely resembles charge spectra obtained from these measurements (we note that the donor hole signature is also present in the PTB7-Th:Y6 measurement, which must be accounted for when making this comparison). We note that the charge is also likely to have a very similar spectrum to an intraspecies charge-transfer[36] state/excimer/weakly-bound exciton delocalised over 2 or more molecules in the Y6 crystal, and therefore designate the spectra here as representing the free charge plus CT state populations. We stress that we cannot explain all of our data unless there is a significant fraction of free charges, as detailed below.

We confirm that the blue species is that of the singlet exciton in 2 ways: (1) through TA measurements of a 'solid-solution' of Y6 diluted in polystyrene (Supplementary Fig. 2), and (2) by comparing the TA spectra of the 'blue' species to the transient kinetics of upconverted PL signal from the same film at roughly similar excitation density. As shown in Fig. 2c, the singlet TA matches the transient PL signal. The singlet, and free charge/CT species show different kinetic decays to each other across multiple different excitation density regimes (see below). As an initial indication of bimolecularity, we note that the square of the charge population closely matches the derivative of the charge, even at low fluence.

The singlet and charge species, as shown in Fig. 2d, show a prompt, fluence independent (Supplementary Fig. 4), interconversion within the first 2.5 picoseconds, which also has no clear excitation-energy dependence (observable within our excitation energy resolution – Supplementary Fig. 5). These observations are consistent with observations from Wang et al.[11], though our other measurements (terahertz spectroscopy, ultrafast transient PL, and fluence dependent PLQE) necessitate a model beyond pure CT generation. As this effect appears not to be due to excess energy (based on TA measurements pumped with different excess energies), the rapid formation is likely an effect of increased exciton delocalisation at early times[37,38]. In line with Gillet et al.[39], we also observe a small 3rd spectral component in the TA which is consistent with a triplet signature (Supplementary Fig. 6). As further evidence that there are significant free charges present, the triplet signal peak is significantly delayed compared to the charge/CT signal, which is consistent with a non-geminate triplet generation pathway rather than geminate triplet formation from inter-system crossing. Additionally, optical-pump-terahertz-probe spectroscopy (Supplementary Fig. 17) shows a conductivity spectrum consistent with a fraction of free charges present at ~3 ps pump delay time (although we note that this spectrum could also be consistent with a pure CT-state population – insufficient signal-noise ratio in the measurement prevents more precise interpretation).

We fit our transient absorption kinetics and intensity-dependent PLQE data based on a basic kinetic model of free charges and excitons present in a neat film of Y6 upon

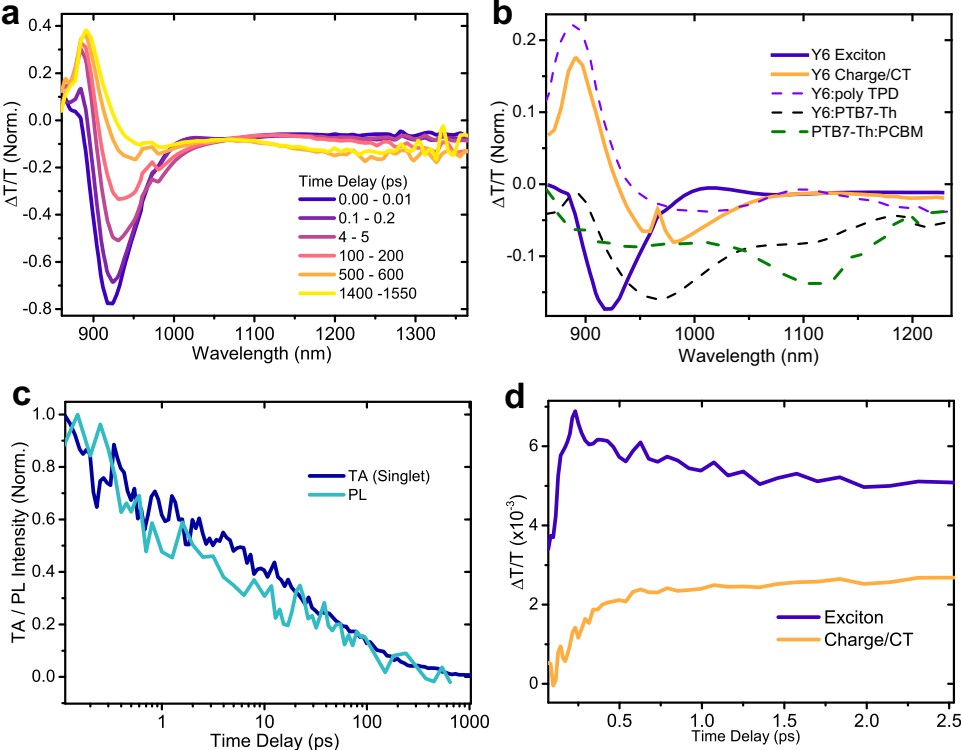

**Fig. 2 Ultrafast transient absorption spectra and kinetics of singlets and free charges in neat Y6, and optical pump terahertz probe conductivity.**
**a** Transient absorption spectra, normalised to the absolute area integral, of a neat film of Y6 at different time delays. 700 nm, 200 fs pump pulse, excitation density ~ $10^{16}$ excitations/cm$^3$. The spectrum evolves from the exciton, blue line, with a negative peak at 915 nm, to the red-shifted charge/CT state spectrum, orange line. **b** Confirmation of charge and exciton transient absorption spectra by comparison of neat species (extracted using genetic algorithm) with species resolved from blends of Y6 with hole accepting materials (PTB7-Th, polyTPD). The green dashed line shows the hole spectra of PTB7-Th blended with PCBM. **c** Singlet component from GA of transient absorption (dark blue line), at an excitation density of ~$8 \times 10^{16}$ cm$^{-3}$, compared with up-converted PL signal (cyan), at an excitation density of ~$2 \times 10^{16}$ cm$^{-3}$. **d** Initial kinetics of the exciton (blue) versus charge/CT (orange) TA kinetics in the first 2.5 ps after photoexcitation. The charge species shows a rise concomitant with the fast initial decay of the singlet species. 550 nm, 200 fs pump pulse, excitation density ~ $10^{18}$ excitations/cm$^3$.

illumination. Fig. 3a graphically represents this model (with further details in SI and Fig. 3b, c). The key processes illustrated and modelled are: Photogeneration of an initial, delocalised singlet exciton, $S_1^*$, which rapidly evolves into localised excitons, $S_1$, and free electrons and holes, $FC$. Free charges bimolecularly recombine to form singlets and triplets, or non-radiatively via trap/defect states. This basic model is all that is required to accurately recreate our experimental data. We can optionally include further processes, such as triplet-triplet annihilation, or specify the traps as most likely being hole traps[40], to improve our fits to the data further. Our data fails to fit to a model that explicitly includes a substantial and long-lived CT state population (see SI for further details).

Figure 3b–d show the results of a global fit of this basic model to a multiple fluence series of transient absorption, and PLQE data (with TTA and trap effects also shown in Fig. 3d). We calculate internal PLQE[41] (taking photon recycling into account – see below) as the ratio of radiative decay events to total decay events, integrated over a microsecond, with a simulated 600 ps input photon pulse. The radiative and non-radiative singlet decay rates are obtained from comparison of the PLQE values of Y6 in a solid-state solution of polystyrene.

These models simultaneously fit the transient absorption data, and crucially, fit the large increase, or 'hump', in PLQE with only a small number of free parameters (as low as 4). The fitted parameters correspond to physically meaningful rate constants in agreement with literature. We thus view them as instructive for estimating free charge yields and transition rates in Y6. Error

analysis of these models is shown in the Supplementary Information. Though there is interdependency between the fitted rates, and hence the individual error in each rate constant is significant (and model dependent), we can use these models to put a lower and upper bound on steady-state free charge yields at given excitation intensities. Setting the conservative criterion that there is a 10% rise in PLQE with excitation intensity, the models show that the exciton dissociation probability (the proportion of excitons that dissociate to charges within their lifetime) lies in the range 0.6–0.9, and the corresponding steady-state free charge fraction lies in the range 0.65–0.99 at 1 Sun excitation intensity. Our total carrier density under AM1.5 lies in the range $10^{13}$–$10^{14}$ cm$^{-3}$ depending on the model used.

The red line in Fig. 3e illustrates the fraction of steady-state free charges present based on a basic kinetic model with no TTA or hole trap filling. Almost completely overlapping this line is a blue curve given by the Saha equation, with an excitonic binding energy of 270 meV (corresponding to an exciton effective mass of ~0.5 $m_e$). The very close match between the two curves shows the equivalence of the kinetic model to the Saha equation.

**High photon reabsorption in thin films.** As illustrated in Fig. 3d, it is important to take into account photon reabsorption in PLQE measurements of Y6 thin films. This is a further consequence of their extremely high refractive index. We use the same process as Richter et al.[41] to approximate photon reabsorption, and show that the true PLQE of Y6 is much higher (~6%) than what would

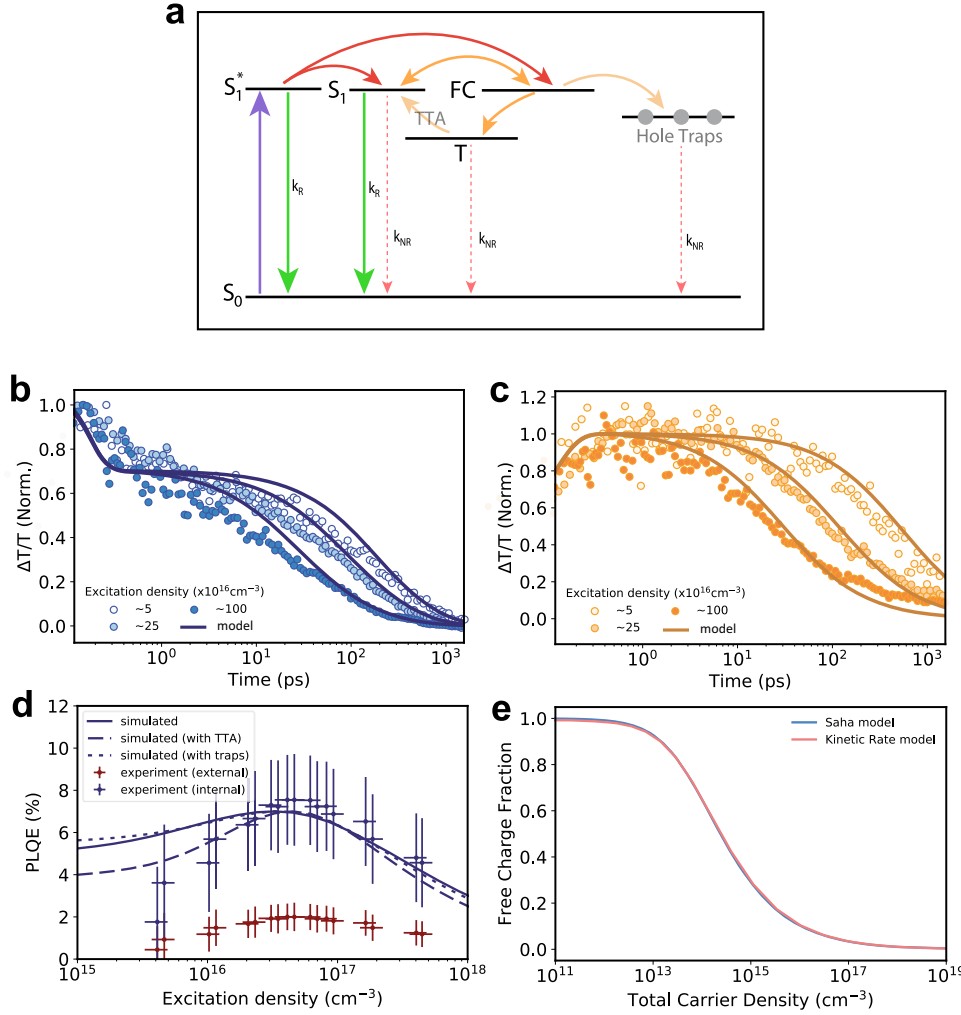

**Fig. 3 Kinetic model of charge generation in Y6 with transient absorption, PLQE and predicted steady-state free charge fractions. a** Jablonski diagram highlighting key excited-state pathways – rapid initial interconversion (solid red arrows), charge recombination (solid orange arrows), non-radiative (red dashed arrows), and radiative rates (green arrows). **b** Normalised transient absorption kinetics of excitons (blue circles) and (**c**) charge states (orange circles) at different excitation densities, fitted with a global fit to the basic kinetic model described in the text and SI (solid blue and orange lines). **d** Internal (blue crosses) and external (red crosses) PLQE values of Y6 as a function of excitation density, with corresponding simulated values from fits to the transient absorption and intensity dependent PLQE. Internal PLQE is calculated from external PLQE as per Richter et al.[41] The solid blue line is the simulated PLQE from the basic model, the dashed blue line shows the PLQE calculated when TTA is included, and the dotted blue line shows the estimated PLQE from an explicit treatment of hole traps (but no TTA). **e** Steady-state free charge fraction as a function of total excitation density calculated from the rate constants gathered from the basic kinetic model (red line), and from an estimate from the Saha equation (blue line). The Saha equation is calculated based on an excitonic binding energy of 270 meV, corresponding to an exciton effective mass of ~0.5 $m_e$.

be ascertained from a direct reading of the external PLQE (~2%). This has significant importance for solar cell material optimisation efforts, as maximising PLQE is a useful approach for assessing photovoltaic efficiency potential[42], and ultimately, being able to harness radiative carrier recombination through photon recycling is key to approaching the Shockley-Quiesser limit. Although we estimate photon recycling is small for these measurements (Supplementary Fig. 16), the high refractive index means that it will become significant if internal PLQE can be increased.

**Fast charge recombination in single component OPV devices.**
High intrinsic charge yields suggest a new path for OPV design by creating single component devices, or devices with very low donor contents (which may also be beneficial for semi-transparent OPV). However, the results from the transient absorption, PLQE, and terahertz spectroscopy data explain why,

even though charges are intrinsically generated, single component, monolayer devices are not efficient. High bimolecular charge recombination, combined with a significant hole trap population, mean that free charges recombine before arriving at electrodes in a device. With our measured charge recombination rates, and assuming (for the sake of illustration), our monomolecular charge recombination term to give the minority carrier lifetime, the hole diffusion length would be less than 15 nm. We fabricated single component Y6 devices. The highest PCE based on a ITO/PEDOT:PSS/Y6/LiF/Al structure was 0.09%. By replacing PEDOT:PSS with PCP-Na, whose HOMO better aligns with that of Y6, the highest PCE obtained was 0.63% (Supplementary Fig. 7). This value is low (though there is no junction in the active layer to separate charges and prevent minority carrier recombination). This loss of photocurrent and photovoltage is due in part to high bimolecular, and space-charge induced recombination, which we show by performing intensity dependent short-circuit

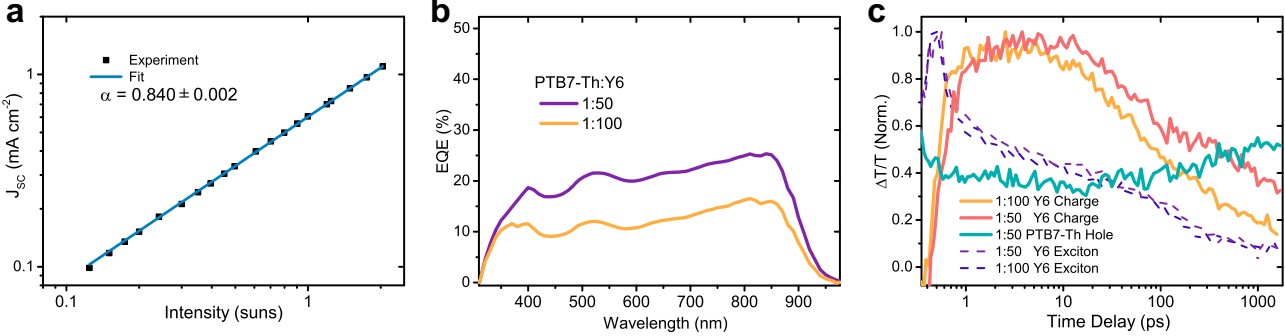

**Fig. 4 Single component short circuit current density, low donor active layer device EQEs, and corresponding exciton and charge kinetics. a** Intensity dependent $J_{SC}$ of a monolayer Y6 device. The power exponent fitted to the data between 0.2 and 2 suns is 0.84, indicating a high number of bimolecular recombination events. **b** External quantum efficiencies (EQEs) of devices with active layers consisting of small ratios of donor, PTB7-Th, to acceptor, Y6. For such low donor content, EQEs are unexpectedly high. **c** Normalised transient absorption kinetics of exciton and charge species in blends of PTB7-Th:Y6 with ratios of 1:50 and 1:100. Pumped with 800 nm, 150 fs pulses at excitation density $1 \times 10^{17}$ excitations/cm$^3$. The exciton kinetics of both films, dashed lines, are matching, as are the rise times of the charge kinetics (orange and red lines), but the 1:50 blend shows slower charge recombination after 20 ps. The green-blue line shows the hole kinetics in the 1:50 blend, showing a gradual increase in hole population as excitons and charges in Y6 diffuse to the PTB7-Th.

current ($J_{SC}$) measurements. Previous reports have seen power law dependence of $J_{SC}$ on illumination intensity ($I$) according to:[43,44] $J_{SC} \propto I^{\alpha}$. Deviations of the fitting parameter, $\alpha$, from unity signify bimolecular recombination or space-charge effects due to unbalanced charge mobilities[43–46]. Electron and hole mobilities are relatively balanced in Y6 (being $1.8 - 40 \times 10^{-4}$ and $0.5 - 50 \times 10^{-3}$ cm$^2$ V$^{-1}$ s$^{-1}$ respectively[24,47]), and at intensities below 0.01 suns, the exponent, $\alpha$, approaches 1 (Supplementary Fig. 8). Fig. 4a shows that for the region between 0.2 and 2 suns incident photo-intensity, the fitted exponent is $\alpha = 0.84$, indicating the device suffers from significant bimolecular recombination and/or space-charge effects at 1 sun.

To further emphasise the importance of high charge recombination, rather than exciton splitting, as a limiting factor in device efficiency, we investigated the effect of 'doping' our Y6 material with very small amounts of donor polymer, PTB7-Th[48]. These very dilute blends allow us to temporally separate the processes of charge separation, transport, and recombination in transient absorption measurements. We measured the photovoltaic external quantum efficiencies of the corresponding devices, which both show surprisingly high efficiencies. As shown in Fig. 4b, when blended in a 1:50 ratio of PTB7-Th:Y6, the EQE is nearly twice as high as for the 1:100 parts blend. However, Fig. 4c shows that this is not due to differences in exciton splitting. The transient absorption kinetics show that for both blends the exciton species decay, and the charge species appear, with almost identical kinetics in the first 20 ps, but the charges recombine faster in the 1:100 blend, showing that the polymer donor's key role is to decrease the rate of recombination. For both bimolecular charge recombination, and hole-trap (or n-dopant) induced recombination, once holes are removed from the Y6 - as they are, rapidly, in normal bulk heterojunction (BHJ) blends (as shown in Supplementary Fig. 10) - then the left-over electrons have nothing with which to recombine, and their recombination kinetics are slowed after electron and hole separation.

**Molecular picture of charge generation.** Density functional theory calculations were used to gain an understanding of the structural features responsible for free charge generation in Y6. The first requirement is significant electronic coupling between $S_1$ and $CT_1$. This parameter was calculated for all molecular pairs exhibiting π-π stacking interactions in the Y6 crystal structure[24] by projecting the quasi-diabatic exciton/CT states calculated for a

pair of separated molecules onto the adiabatic states of the molecular pair in the crystal geometry[49,50]. This approach enables us to simultaneously predict the energies and couplings for exciton/CT states (Fig. 5a, Supplementary Fig. 22, and Supplementary Table 1).

The computed couplings and energetics point to a kinetic pathway for exciton break-up. The exciton-CT coupling is significant (10–75 meV), with the CT states either equal or lower in energy to the excitonic states (except one pair in which the stacking interaction is minimal). These two factors, combined with the low reorganisation energy (estimated as ~0.3 eV based on the individual molecules) imply that exciton dissociation is expected, even according to incoherent charge transfer theories. The same electronic coupling parameters/state energies are also consistent with the possibility of coherent CS, as we observe spectroscopically. Indeed, as previously noted, the coupling between excitonic states suggests a high degree of exciton delocalisation[24], which is expected to lower the reorganisation barrier for charge formation. Coupling of the CT states with the ground state is also strong (30–85 meV), which is consistent with the rapid recombination observed experimentally. We note that the CT-GS coupling strengths do not correlate with the exciton-CT coupling in a simple manner. This apparent independence of $V_{Ex-CT}$ and $V_{CT-GS}$ indicates that the recombination can, at least in principle, be overcome by crystal structure engineering to maximise the $V_{Ex-CT}/V_{CT-GS}$ ratio. Evidence of exciton/CT hybridisation may be seen in the broader red tail of the UV-vis absorption spectra.

Our calculations also offer a clue regarding the origin of charge formation in Y6. In pairs involving molecules from different lattice positions, there is a clear energetic preference (~0.15–0.3 eV) for which of the two possible CT states is formed. These observations indicate that the packing geometry results in two distinct polarisation environments, creating a donor-acceptor system through supramolecular, rather than synthetic means. This interpretation also explains the significant recombination observed experimentally: the alternation of 'donor' and 'acceptor' positions on the scale of the unit cell is akin to an extremely intermixed blend system – meaning that both charge formation and recombination are extremely rapid.

The energy levels calculated for free charges also reveal the presence of distinct polarisation environments. Fig. 5b shows the ionisation energy (IE) distributions calculated for a Y6 thin film using the long-range electrostatic embedding procedure available

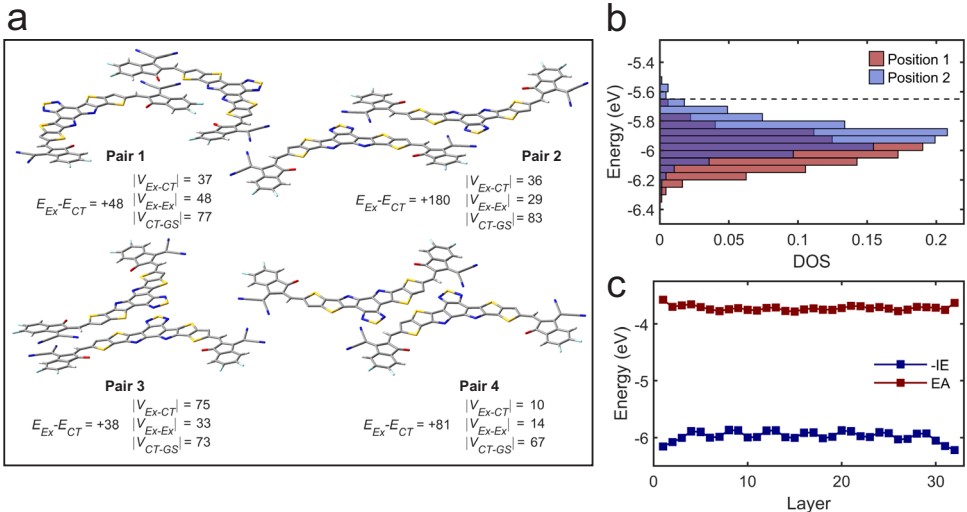

**Fig. 5 Electronic couplings and energies for Y6 excitons, CT states, and charges. a** Exciton-CT energy offsets and electronic couplings (in meV) for selected π-stacked molecular pairs extracted from the Y6 crystal structure (alkyl chains have been removed for clarity, remaining structures can be found in the Supporting Information)[24]. **b** Density of states (DOS) for holes (−IE) in a 10 nm thick model thin film based on molecular dynamics equilibration of the Y6 crystal structure, normalized to unit area. Dashed line indicates the DOS onset determined by photoelectron spectroscopy in air[6]. **c** Average ionisation energy as a function of π-stacking layer showing differential hole stabilisation for the different lattice positions, which occur in an "...ABBA..." sequence.

in VOTCA (electron affinity (EA) is shown in Supplementary Fig. 13)[51–57]. Figure 5c shows that the different energetics of charges occupying distinct crystal positions results in a bimodal density of states, particularly for holes. These distributions confirm the presence of a driving force for charge formation inherent to the Y6 packing structure, on the order of the exciton binding energy. This situation bears some similarity to single component OPVs in which crystalline domains with different orientations possess distinct charge energetics, resulting in charge formation at interfacial boundaries[58]. The present work is distinguished from the prior literature in that the different energetics are inherent to the crystal structure, resulting in bulk - rather than interfacial - charge generation.

In summary, our calculations reveal equal energy exciton-CT manifolds (as a result of differential polarisation), strong exciton-CT coupling, and a flat energetic landscape for electron transport. Taken together with previous theoretical work showing appreciable electronic coupling for electron and hole transfer, these observations strongly suggest the existence of a kinetic pathway for exciton-to-free charge conversion in Y6.

## Discussion

The observation of intrinsic charge generation in a single organic material necessitates an expanded understanding of the photovoltaic process in organic materials. The implications for device design are far-reaching, showing that a new approach to reaching maximum efficiency may be to maximise intrinsic charge generation yield, whilst reducing rapid trap/dopant-assisted monomolecular and bimolecular charge recombination, rather than focus on exciton splitting in a donor-acceptor blend. Indeed, we suggest that the highest efficiency OPV blends (such as PM6:Y6) have in fact been optimised for the extraction of intrinsically formed charges.

We propose a mechanism for how recombination is suppressed in current Y6 blends, drawing on studies invoking energy level bending at the donor/acceptor interface as a result of long-range interactions of charges with molecular quadrupole moments[6,22,51,59–61]. In an excitonic picture – where charges only form at the interface - these

fields assist CT dissociation and suppress non-geminate recombination. However, these benefits come at the expense of exciton dissociation rates, resulting in an intrinsic limit for the ionisation energy offset between the neat materials[6]. When free charges are intrinsically generated in Y6, not only is the interfacial exciton dissociation barrier bypassed, but the level bending defines an attractive potential for holes, which are pulled towards the interface before leaving the acceptor material, while electrons are repelled. Under this interpretation, champion efficiency systems, such as PM6/Y6, benefit from both exciton- and charge-funnelling effects: excitons in PM6 are funnelled to Y6 via resonant energy transfer[6,27], while level bending pulls holes from Y6 into the polymer domain, and suppresses recombination thereafter. We can test whether quadrupolar fields affect charge recombination in Y6 by comparing charge and exciton kinetics of neat Y6, to a film of Y6 blended with 20% weight ratio PCBM (a ratio present in highly efficient ternary devices[28]). The energy levels of PCBM mean that both electron and hole transfer from Y6 to PCBM are prohibited. Any reduction in charge recombination rate will therefore be due to interfacial level bending (one would expect the effect of increased crystallinity in Y6 due to the PCBM would increase recombination, rather than decrease). Supplementary Fig. 11 shows that, as predicted, there is a small reduction in charge recombination rate, and concurrent reduction of exciton population, when PCBM is introduced.

Efficient intrinsic charge generation, in the FREA small molecule Y6, has broad implications for OPV device design. While previous works (on materials such as P3HT[62] and PCBM[63]) have generated debate, showing evidence of a proportion of (10–15%) charges generated in neat films, this work differs significantly in that the free charges are not formed 'extrinsically' (in the case of P3HT[64]), or from field-induced exciton separation[10]. Considering a population of excitons and free charges in neat Y6 necessitates a more nuanced understanding of the simplified four step process – exciton generation, exciton transport, charge separation at a heterojunction, charge transport – that has previously dominated OPV discussion. Exciton diffusion length is no longer a simple function of how fast a lone exciton travels and how long it lives - interconversion into free charges and the reverse process must also be considered. The ease

of charge separation, but concurrent rapid charge recombination, necessitate a renewed focus on reducing such recombination. This hints that for current Y6-based devices the key purpose of the bulk heterojunction is more to reduce charge recombination, than to split excitons. Emphasis may shift from not only reducing interfacial and CT state bimolecular recombination, but to also focussing on a reduction in minority-carrier recombination. Simultaneously maximising intrinsic PLQE and charge generation should improve OPV device open circuit voltages – which have yet to achieve parity with inorganic photovoltaics. Specific directions of study include: improving crystal packing to further enhance dielectric constants and hence free charge fraction; tuning energy levels through packing structures and/or molecular orientation rather than different compounds; and crystal structure engineering to minimise CT-ground state coupling.

We have shown, through intensity-dependent PLQE measurements, that free charges exist in neat films of the molecular semiconductor Y6, at illumination intensities below 1 sun. We have corroborated this measurement with optically-pumped time-resolved terahertz, ultrafast photoluminescence up-conversion and transient absorption spectroscopy. Our data can be explained by simple kinetic models, along with a new quantum mechanical picture of exciton-CT state coupling and free charge energetics. We have outlined the implications of this finding for our understanding of the photovoltaic process, and the mechanism of action for current champion efficiency devices, and have suggested further avenues of study for enhancing efficiency in future devices. These findings open up the potential to think beyond the bulk heterojunction, and revisit the possibility of fabricating efficient doped organic p-n junctions.

## Methods

**General**. UV-Vis absorption spectra were obtained using Carry 5000 (Agilent) and a Cary 50 Bio UV-vis spectrometer in the range 190–1100 nm. Photoluminescence spectra were obtained using a Cary Eclipse (Varian). Photoluminescence quantum efficiency measurements were obtained in an integrating sphere using the method of de Mello et al.[65]. A Q-switched frequency doubled ND:YAg laser with output wavelength 532 nm, pulse-length 600 ps, and rep-rate set to 25 kHz was used to excite the sample for the intensity dependent PLQE measurements. Sample film thicknesses were performed using a Dektak profilometer. Thin films of Y6, and Y6 blends were spin-coated onto quartz spectrosil substrates in a glovebox from 11 mg/ml chloroform solutions, at a spin-speed of 3000 rpm. Solid solutions of Y6:polystyrene were prepared in a 1:50 weight ratio, and spincoated in a glovebox at 800 rpm.

**Device fabrication**. Indium tin oxide-patterned glass substrates were cleaned by sequential sonication in acetone and propan-2-ol and subjected to ozone treatment. A 35 nm film of PEDOT:PSS was spin coated in air from aqueous solution and baked on a hotplate at 110 °C for 15 min. A 30 nm layer of Y6 was spin coated from a chlorobenzene solution in a nitrogen-filled glovebox (0.1 ppm $O_2$, 0.0 ppm $H_2O$). The cathode of 1 nm LiF and 90 nm Al was deposited by vacuum thermal evaporation at pressure ~$10^{-6}$ mbar through a shadow mask to define mm$^2$ pixels. The devices based on PTB7-Th:Y6 were fabricated with a normal structure as ITO glass/PEDOT:PSS/PTB7-Th:Y6/PNDIT-F3N/Ag. Patterned ITO glass was pre-cleaned in an ultrasonic bath with deionized water, acetone and isopropanol, and treated in an ultraviolet–ozone chamber (Jelight Company, USA) for 20 min. PEDOT:PSS was spin-coated on the pre-cleaned ITO at 5000 rpm, followed by baking at 150 °C for 15 min. Then, the active layers were spin-coated on PEDOT:PSS (PTB7-Th:Y6, X:12 mg mL$^{-1}$ in chloroform, 2500 rpm). Afterwards, the PNDIT-F3N solution (0.5 mg mL$^{-1}$ in methanol) was spin-coated on the active layer at 2000 rpm. Finally, Ag electrode (ca. 80 nm) was slowly evaporated onto the surface of the underneath layer under vacuum (ca. 10$^{-5}$ Pa). The devices were not masked and the active area of devices were 4 mm$^2$.

**Device characterisation**. Photovoltaic device JV characteristics were measured in a nitrogen-filled glove box at room temperature using a solar simulator and source-measure unit. Light intensity was determined using a calibrated silicon diode. We tested 20 devices for each kind and choose the best performing devices. Light intensity was varied by adjusting the distance between source and device under test and also by use of neutral density filters. EQE spectra of PTB7-Th:Y6 devices were obtained using a Solar Cell Spectral Response Measurement System QE-R3011 (Enlitech Co.).

**Transient absorption spectroscopy**. Measurements were performed using a homebuilt experimental setup illuminated by an amplified Ti:sapphire laser, with pulse durations of 100–150 fs, centred around 800 nm and at a repetition rate of 3 kHz. The excitation pulses are generated either using this fundamental, or by using an optical parametric amplifier (TOPAS) with the 800 nm fundamental input and then chopped at $f/2$ (1.5 kHz). Photoexcitations in the materials were probed via a broadband white light continuum generated by focusing a portion of the fundamental to an undoped YAG (Yttrium Aluminium Garnet) crystal. Pump-probe polarizations were kept under magic angle (54.7$^0$) configuration to avoid orientational dynamics. After passing through the photoexcited sample, the probe pulses were spectrally dispersed using a prism spectrometer and are then collected using a CMOS camera (visible components) or an InGaAs photodiode array (IR components). The time resolution is obtained via introducing time delays in the pump path which is achieved using a retroreflector connected to a motorized translational stage. The differential transmission signals at various time delays are calculated from the sequential probe shots corresponding to the pump on versus off. For typical measurements, 8000 shots were averaged at each time point and were repeated at least four times. To avoid degradation, all the samples were measured under a vacuum, or inert nitrogen environment.

**Time-resolved terahertz spectroscopy**. Optical pump terahertz probe spectroscopy was performed using a dual lock-in technique, similar to that described by Tiwana et al.[66]. THz pulses are generated by an amplified Ti:sapphire laser, with pulse durations of 100–150 fs, centred around 800 nm, generated at a repetition rate of 3 kHz chopped to 1.5 kHz, incident onto a 1 mm thick ZnTe crystal. THz pulses are measured by balanced photodetectors using electrooptic sampling in a separate ZnTe crystal. The 800 nm 'optical' pump is chopped at 750 Hz, and incident on the sample with a spot-size diameter of 4–5 mm. The THz probe diameter was measured as ~1–2 mm. The entire THz beam path is enclosed in a cavity pumped with dry air, with humidity monitored to ensure % water remains below 2%. The sample is enclosed in a nitrogen environment to avoid laser degradation and is excited from the glass substrate side. THz conductivity was calculated[66] as:

$$\triangle\sigma = -\frac{\varepsilon_0 c(n_A + n_B)}{L}\frac{\triangle T}{T}$$

where $\varepsilon_0$ is the vacuum permittivity, $c$ is the speed of light, $n_A$ and $n_B$ are the refractive indices of the media on both sides of the sample in the THz region – quartz, and air, $L$ is the sample thickness, and $\frac{\triangle T}{T}$ the measured change in THz transmittance.

**Ultrafast transient photoluminescence spectroscopy**. Ultrafast photo-luminescence dynamics were measured using a photoluminescence up-conversion technique. The setup's light source is a Ytterbium fiber laser (Tangerine SP, Amplitude Systemes) operating at 44 kHz and generating 150 fs pulses. The laser output was split into pump and gate parts. The pump was frequency-doubled to 515 nm using a BBO crystal and focused to a 50 µm$^2$ spot to excite the sample. The sample fluorescence was collimated and refocused on a nonlinear mixing crystal (1 mm BBO, θ = 32°) by a pair of aluminium-coated off-axis parabolic mirrors. The crystal angle is set for the phase-matching condition (type I) to produce sum frequency signals at 480 nm from mixing the 1030 nm gate and 900 nm photo-luminescence. Two achromatic lenses collimated and focussed the upconverted signals onto the spectrometer (Princeton Instruments SP 2150). Signals were detected by an intensified CCD camera (Princeton Instruments, PIMAX3). A combination of 550 nm long- and 800 nm short-pass filters were used to remove residual excitation and 1030 nm gate light, respectively. The time delay between pump and gate beams was controlled via a motorized optical delay line on the excitation beam path. For each delay time, 240,000 shots were accumulated. Samples were measured under a vacuum environment.

**Density functional theory calculations**. Density functional theory calculations were performed using Gaussian 09[67] with an ultrafine integration grid, and the accuracy of two-electron integrals increased to 10$^{11}$. Optimized geometries were confirmed by calculation of the associated vibrational modes, which revealed no imaginary frequencies in all cases.

**Electronic coupling calculations**. Electronic coupling calculations were conducted on pairs of molecules extracted from the X-ray crystal structure, with alkyl chains truncated to ethyl. These calculations used the long-range corrected CAM-B3LYP[68] exchange–correlation functional in combination with the 6–31+G(d,p) basis set. Linear response TD-DFT calculations for excited states were performed within the Tamm–Dankoff approximation (TDA). Dielectric stabilisation was treated using a polarisable continuum model with $\epsilon = 5.0$, based on the experimentally-determined value[26,69–72]. To treat CT states, the exchange–correlation functional was range-tuned according to the established non-empirical procedure[69–72], resulting in an optimal range-separation parameter of 0.010 Bohr$^{-1}$. Electronic couplings and exciton/CT energetics were accessed by performing calculations in which the molecules were separated from one another by 10 Å to define localized/quasi-diabatic states. At this distance, interactions

involving orbital overlap are negligible, resulting in the formation of localized CT states, and pure exciton states that can be localized by diabatization[73]. The energies and couplings between these states were calculated by projection onto the states of the molecular pair in the crystal packing geometry[49,50].

**Density of states calculations**. Density of states calculations for electrons and holes were performed using the long-range embedding procedure available in VOTCA-CTP[51–57]. Molecular charge densities and polarizabilities for the ground state, cation, and anion were calculated using B3LYP/6–311 G(d,p). The charge densities were used to fit distributed atomic multipoles (specifically, atomic charges, dipoles, and quadrupoles) using GDMA (version 2.3)[74]. Polarizabilities for atoms in alkyl chains were taken directly from the set of Thole polarizabilities[73]. Atomic polarizabilities for atoms in the Y6 chromophore were obtained by scaling Thole polarizabilities to match the DFT-calculated polarisable volume of the core[51]. Solid state corrections to the gas-phase IE/EA were calculated using the long-range Ewald summation scheme in VOTCA-CTP. These calculations employed a molecular dynamics equilibrated $8 \times 8 \times 6$ supercell constructed from the Y6 crystal structure[24], with the π-π stacking direction aligned with the $z$-axis, to match the experimentally observed face-on packing preference relative to the substrate. This simulation cell was used to mimic a ~10 nm thin film by including a ~20 nm vacuum buffer in the $z$ direction and applying the shape term for a periodically repeated supercell in the xy plane. The interaction cut-offs for electrostatic and polarization interactions were 8 and 6 nm, respectively.

**Molecular dynamics simulations**. Molecular dynamics simulations were conducted using the Y6 crystal structure with the missing atoms of the alkyl side chains built in. The system containing $8 \times 8 \times 6$ unit cells was generated using Maestro[75] in the Schrödinger Material Science Suite, and then prepared for molecular dynamics simulation with the OPLS3e force field using the System Builder in Desmond[76–78]. The prepared system first underwent 100 ps of Brownian minimization, followed by 1 ns of equilibration using $NPT$ molecular dynamics simulation at 1 fs timestep, 300 K and 1.01325 bar. A further 10 ps molecular dynamics simulation was conducted to collect 200 frames of trajectories at 50 fs time interval for analysis.

**Reporting summary**. Further information on research design is available in the Nature Research Reporting Summary linked to this article.

## Data availability

The raw data and code for data processing can be found at https://github.com/Mikebprice/Free-charge-generation-in-Y6. Questions regarding this data set are encouraged to be sent to the corresponding authors.

## Code availability

Code that was used to support the findings of this study (and which has not been previously made available) can be accessed from https://github.com/PaulAlexanderHume/Free-charge-generation-in-Y6-theory. Questions regarding this data set are encouraged to be sent to the corresponding authors.

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

## Acknowledgements

M.B.P., P.A.H., A.I., I.W., R.R.T., K.E.T., N.J.L.K.D., K.C. and J.M.H. acknowledge support from the MacDiarmid Institute of New Zealand. P.A.H., A.I., I.W., R.R.T., K.E.T., N.J.L.K.D., K.C. and J.M.H. acknowledge funding from the Marsden Fund of NZ. M.B.P. and N.J.L.K.D. acknowledge support from the Royal Society of New Zealand. A.G, P.J.C. and G.L. acknowledge support from ARC Centre of Excellence in Exciton Science (funding grant number CE170100026). P.A.H. would like to acknowledge the support of Professor Denis Andrienko, Dr Anastasia Markina and Dr. Carl Pölking for support in utilising VOTCA software. All calculations were performed using the VUW High Performance ComputingCluster "Rāpoi".

## Author contributions

M.B.P., P.A.H.: conceptualisation, investigation, experiments, supervision, writing. M.B.P., P.A.H. contributed equally. A.I., I.W., R.R.T., K.E.T., A.G., P.J.C., N.J.L.K.D., K.C., P.X., H.L.: experiments, investigation. W.J.: molecular dynamics simulations. G.L.: supervision. Y.W.: supervision, investigation, conceptualisation, X.Z.: supervision, conceptualisation, funding acquisition, J.M.H.: conceptualisation, investigation, supervision, writing, funding acquisition.

## Competing interests

The authors declare no competing interests.

## Additional information

**Editorial Note** This manuscript was previously reviewed at another journal within the Nature Portfolio. This document only contains reviewer comments and rebuttal letters for versions considered at Nature Communications.

