## [Peer Review File · Nature Communications]

Free Charge Photogeneration in a Single Component High Photovoltaic Efficiency Organic SemiconductorEditorial Note: This manuscript was previously reviewed at another journal within the Nature Portfolio. This document only contains reviewer comments and rebuttal letters for versions considered at Nature Communications.

REVIEWER COMMENTS

Reviewer #1 (Remarks to the Author):

Hodgkiss et al. have claimed the observation of high efficient free charge generation upon optical excitation in the film of small molecule Y6. The paper has previously been reviewed by three referees. In the authors' response, they have rebutted the criticism on the novelty. In my opinion, the manuscript if valid provides new understanding about Y6 — the state-of-art acceptors of technical significance although I don't go all the way with the author. I'd recommend publishing the paper in Nature Communications if the authors can strengthen the conclusion.

1) The conclusion of free charge generation is reached based on model analysis. While the data of different spectroscopic measurements provide a self-consistent story that can be described by assuming the scenario of exciton-charge equilibrium, the smoking gun of free charge generation is not reported. The argument is not more convincing than that in the literature (Li et al. *Angew. Chem.* 2021, 60, 15348) as the authors criticized where the thermal activation of exciton dissociation is assumed to be responsible for temperature-dependent PL emission. I'd suggest the author to include the TA data covering the visible range and compare the spectrum with the spectral feature of polarons. The polaron features can be obtained by spectro-electrochemical measurements. The results may also aid to quantify the yield of free charge generation.

2) THz data were reported to support the formation of free charges (Figure S9). Nevertheless, the author noted that the THz spectrum can be interpreted by either the charge formation or excitonic polarizability. In principle, the contribution from the two excitation species may be distinguished by further analysis of the THz response spectrum, i.e., the traces probed at the maximum field change or zero crossing (e.g., Kumar et al., *Sci. Adv.* 2020, 6, eaax8221). These analyses may provide more convincing results to justify the free charges.

3) Saha model (Equation 1) has been adopted as the major tool to interpret the experimental data. Saha model was originally developed assuming thermodynamic equilibrium. Simply, this is not valid at all the temporal stages for ultrafast transient optical measurements. I suggest the authors to comment on this issue and clarify the scale when the model is valid.

Reviewer #2 (Remarks to the Author):

The authors have reported intrinsic charge generation in Y6 molecule, which is one of the most popular OPV materials currently. Both spectroscopic analysis and theoretical calculation point towards the splitting of excitons in the bulk of Y6, without a molecular heterojunction. I first of all agree with other reviewers that generation of free charges in single component D-A type organic film is not new, but I do find the experimental results of this work fundamentally interesting and can potentially attract broad readership from optoelectronic community, as it offers alternative path for potential new type of optoelectronics.

The authors have improved the technical part as suggested by other reviewers, but I also agree with other reviewers the impact of this work is not properly revealed. I don't feel the data authors presented in latter part of this work actually demonstrated helpful insight to either explain currently Y6-based highly efficient OPV/to guide future optimization of OPV, nor do I feel the current results challenge mechanism of current Y6 based OPV, as written in the abstract. At this stage, I can't recommend to accept this manuscript in *Nat. Commun.* without major revision. Some data needs further clarifying and the way authors deliver the insight in latter part of this work needs to be re-shaped. Some questions and

suggestions below perhaps to help improving the quality of the manuscript.

1. The author claims the mechanism of free charge generation in Y6 film is due to strong coupling between exciton and CT state, I would therefore expect a sort of hybridization of these two signatures in TA, perhaps forming a new signature. As a matter of fact, the charge signature of Y6 pristine film show the same pattern as compared to that in Y6:poly-TPD in figure 2b, the authors need to comment on this phenomenon (is it possible to directly excite the CT state?). Additionally, can authors comment on the effect of excitation energy upon charge yield? Figure S7 shows the exciton kinetics doesn't change upon excitation energy. Once excitons are coupled with CT states to generate charges, it is hard to believe excitation energy doesn't affect this process at all (see *Polymer*, 52 (2011), 4397-4417).

2. By looking at the charge kinetics in figure 3c, the lifetime of charges seems to be relatively short (averagely around 100-200 ps). The authors claim there is trap-assisted recombination and bimolecular recombination, and such loss mechanism happens after 100 ps (open circle in figure 3c, presumably in linear region of excitation intensity?). If this is true in BHJ (considering the neat phase), the carrier lifetime in BHJ should also be short, can authors provide similar measurements in normal PM6:Y6 BHJ? It's hard to believe highly efficient Y6-based OPV could suffer from such loss as much as shown in figure 3c. On the other hand, figure 4c shows once a molecular heterojunction is presented, recombination happens as fast as 10-20 ps. It looks like a molecular heterojunction is a "minus" for the survival of free charges here, because there is additional loss mechanism such as geminate recombination (authors need identify the recombination mechanism here)? My question is what can be gained from putting a hole acceptor here if the intrinsic generated charge is as high as 90%?

3. Another question related, as in response to reviewer, authors answered the second question of referee #1, saying "Additionally, while the single component device efficiency is currently low (as would be expected due to not having a p-n junction)". My question is: the recombination mechanism presented in Y6 pristine film quenching free charges should also be presented in BHJ solar cells (geminate recombination in mixing phase and trap/bimolecular recombination in pure Y6 phase), do we even want a p-n junction in terms of keeping charges alive? The "60-90%" free charges generated intrinsically in Y6 film don't survive much beyond 1 ns in either Y6 film or diluted BHJ, I guess a normal BHJ is still the optimal option, as authors said by themselves in page 13, the donor's key role to reduce charge recombination and split excitons. Then what's the insight to be gained from this work in terms of OPV?

4. In general, Y6-based OPV is already highly efficient. The charge generation (exciton to free carrier) is not an issue anymore here. What can be further improved is the non-radiative recombination in the device. Currently, I am not sure how those highly efficient OPV can utilize the larger fraction of intrinsically generated free charge phenomenon here (plus the major trap/bimolecular recombination presented). Additionally, I think the device part is just similar compared to those Schottky junction devices. The authors have done a lot in spectroscopic part, but the device part doesn't prove any insight for the OPV community to push the envelope of device efficiency. I would urge the author to dig more in device part as to give clear information how Y6-based OPV can be further improved. The spectroscopic part alone, otherwise, should go to a more specialized journal.

5. For the single component Y6 device, are authors sure the charge generation mechanism is the same as pristine film without electrode? Once electrodes sandwich Y6, the difference of work function of these metals can provide electric field, which can split intra/inter-molecular CT states, as already reported in other materials. (*J. Phys. Chem. Lett.* 2018, 9, 8, 1885–1892) Perhaps check bias-dependent EQE? Similarly, the dilute BHJ also doesn't prove any new insight how we can utilize the results in spectroscopy, as there are also similar report showing similar effect. (*J. Phys. Chem. Lett.* 2020, 11, 14, 5610–5617) I urge the authors to take a re-consideration how the device part could be presented as to reveal the impact of spectroscopic part.

Reviewer #3 (Remarks to the Author):

In the paper "Free Charge Photogeneration in a Single Component High Photovoltaic Efficiency Organic Semiconductor" the authors show that neat films of the small molecular acceptor Y6 exhibit intrinsic free charges generation. Single component devices built with Y6 have low efficiencies, mainly due to bimolecular charge recombination and intrinsic hole traps. Charge generation is argued to happen due to strong coupling between exciton and CT states and an intermolecular polarization pattern that drives exciton dissociation.

The proposed work concerns an interesting and timely research topic, lately several papers were published on single component OPV but the photophysics in many of them has not been sufficiently investigated, lacking essential information regarding charge carrier dynamics.

The authors provide a very comprehensive study, delivering transient measurements, simulations as well as devices characterization. The main result stems in the simulation, from which it is argued that charges occupying distinct crystal positions, due to packing geometry, have different energetics, resulting in a bimodal density of states particularly for holes. These observations indicate that the packing geometry creates a donor-acceptor system through supramolecular, rather than synthetic means. In a way this result is similar to 10.1038/s41467-020-18439-z where it is proposed that crystalline domains with different orientations generate an electrostatic landscape with an interfacial energy offset of 0.4 eV, which promotes the formation of hybridised exciton/charge-transfer states at the interface, dissociating efficiently into free charges. Unlike this last work, though, in the submitted paper there are no different crystalline domains, rather the different energetics happens at distinct crystal positions. Still, in my opinion this point deserves at least a short discussion in the paper.

Apart from this I only have a small remark:

- pag. 12 it is stated "This value is low (though there is no built-in voltage in the active layer)". I suppose the contacts are ohmic or quasi-ohmic, so at short circuit there is a built in voltage \sim the gap of Y6, I don't understand the reason why there should be no built in voltage

RESPONSE TO REVIEWER COMMENTS

Reviewer #1 (Remarks to the Author):

Hodgkiss et al. have claimed the observation of high efficient free charge generation upon optical excitation in the film of small molecule Y6. The paper has previously been reviewed by three referees. In the authors' response, they have rebutted the criticism on the novelty. In my opinion, the manuscript if valid provides new understanding about Y6 — the state-of-art acceptors of technical significance although I don't go all the way with the author. I'd recommend publishing the paper in Nature Communications if the authors can strengthen the conclusion.

We thank the referee for once again reviewing our manuscript, and noting that the manuscript provides new understanding about Y6, as well as acknowledging the progress and changes made in the manuscript as a response to the previous review rounds. In our below response, we provide additional data as requested by the referee, in the form of the visible part of the TA spectrum, and the charge spectrum gained from chemical doping of Y6, which agree very well with each other, as expected, as well as agreeing very well with the other conclusions in our manuscript. This further strengthens our conclusions regarding free charge generation, which we already considered very strong. We have also modified the manuscript to clear up any parts of the work that were unclear to the reviewer.

1) The conclusion of free charge generation is reached based on model analysis. While the data of different spectroscopic measurements provide a self-consistent story that can be described by assuming the scenario of exciton-charge equilibrium, the smoking gun of free charge generation is not reported. The argument is not more convincing than that in the literature (Li et al. *Angew. Chem.* 2021, 60, 15348) as the authors criticized where the thermal activation of exciton dissociation is assumed to be responsible for temperature-dependent PL emission. I'd suggest the author to include the TA data covering the visible range and compare the spectrum with the spectral feature of polarons. The polaron features can be obtained by spectro-electrochemical measurements. The results may also aid to quantify the yield of free charge generation.

Charge Signature and Quantification. We thank the referee for their concrete suggestions (to provide spectroelectrochemical data and visible TA data) to strengthen the paper's conclusions. Equivalent to the spectroelectrochemical charge signature is the charge signature obtained by chemical doping of a host. This can be performed without requiring specialist spectro-electrochemical equipment that we do not currently have access to. These measurements have in fact been performed in the literature by Wang et al. (<https://pubs.rsc.org/en/content/articlelanding/2019/TA/C9TA10002D>) who chemically doped Y6 with a Lewis acid, and showed that the doped and undoped films had different UV-vis spectra consistent with a p-doped film. We plot these spectra - normalised to the red band edge tail - in the new SI fig S6a (and shown below). When we take the difference of the doped and undoped spectra, we see that the resultant spectrum (orange line) is in very good agreement with the spectra isolated in our paper (Fig2b), as well as being consistent with the visible TA spectra (now also shown in Fig S6b). This chemically-obtained spectrum provides a third independent

confirmation of the charge spectra, alongside that which we obtained through different methods of transient absorption, and transient absorption of dilute solutions in polystyrene combined with ultrafast PL.

While this spectrum provides further confirmation of charge spectral assignment, it alone cannot provide extra information in terms of charge yield quantification, that we do not already have directly from transient absorption. This is the case for spectroelectrochemistry as well. The Di Nuzzo paper (<https://pubs.acs.org/doi/full/10.1021/acs.jpcclett.5b00218>) mentioned by the Referee originally in relation to spectroelectrochemistry does not use the technique to quantify charge (actually polaron-pair) yield, but merely to confirm spectral charge assignment.

We note that we have already comprehensively quantified our charge yield through transient absorption measurements - i.e. by proving what the charge signature is, and then observing it's kinetics under different fluences. This quantification does not rely on any complex model. We have primarily used our simple kinetic model to fit to both the data from transient absorption and intensity-dependent PL, which has enabled us to put error-bounds on our free charge yields, strengthening our conclusions.

Figure S6b shows a representative visible TA spectra, as requested. We didn't initially include this region of the spectrum because it has been published previously more often than the near infrared, is more complex to analyse/deconstruct due to the presence of overlapping stimulated emission, ground-state bleach, and photoinduced absorption signals, and our data in the region from 780-860 nm is less reliable due to the presence of the laser fundamental, and weaker white-light probe. However, we agree with the Referee that it is in fact important to show this data, and it strengthens the paper's conclusions due to its similarity with the charge species.

Figure S1. UV-Vis spectra of Lewis acid-doped films of Y6 showing polaron signature and visible range transient absorption spectra of Y6 films. **a)** UV-vis spectra of undoped Y6 (blue line) and doped Y6 film (pink line) from Wang *et al.*¹¹ normalised to the red tail at 1100 nm. The orange line shows the difference between the two spectra, which gives another representation of charge (polaron) spectra, which matches well with the proposed charge spectra ascertained from transient absorption measurements. **b)** Visible transient absorption spectra of a neat film of Y6, pumped with 550 nm, 200 fs, excitation, at $\sim 5 \times 10^{17}$ excitations/cm³. The spectral region beyond 800 nm is significantly more noisy than other spectral regions due to the presence of the fundamental 800 nm pump, at decreased whitelight probe stability. However the spectral shape matches well with the expected polaron visible features from steady-state spectroscopy shown in Fig S6a.

Relationship to Prior Literature. We believe our argument to be more convincing than the paper mentioned by the Referee (Li et al. *Angew. Chem.* 2021, 60, 15348) for the following reason. In the paper by Li et al., recombination is shown to be thermally activated because the PL efficiency increases at higher temperatures. This behaviour is expected regardless of whether the recombination involves bound CT states, or free charges. Thus, while the paper claims to have proved free charge formation, their data could equally well be explained by a generation of only CT states, and no free charges, in neat Y6. To prove free charge formation, one needs to show that charge recombination follows bimolecular kinetics, because if only CT states are formed, recombination should be monomolecular. Our intensity-dependent PL measurements (and our TA measurements) show that charge recombination is bimolecular, which means that the charges formed in Y6 are free.

We have added a clarifying sentence to the paper to elaborate this point, around line 70:

“In Y6^{19,30} and IDIC,³¹ excitons are delocalised, or form an ‘intra-moiety’ intermediate state with likely charge-transfer (CT) like character. These observations have all been explained by invoking CT-states, however, with additional data, we show here that a more profound explanation is required. To prove that this ‘intra-moiety’ TA signature is in fact due to free charges, we demonstrate that recombination follows bimolecular kinetics, rather than monomolecular decay as would occur for CT states.”

Thus, this proven bimolecularity - which is shown convincingly by our results - (combined with all of our direct spectroscopic evidence of charges) is as close to a smoking gun as it is possible to get, (considering that it is currently infeasible to get THz data at sufficiently low fluences - as we discuss in the next section). Our model has very few free parameters, and we are able to convincingly rule out other explanations based on comparing the results of our transient absorption and intensity-dependent PL. This makes our study substantially different to the Li paper (which we also note once more was submitted after our paper was submitted and the preprint became public).

2) THz data were reported to support the formation of free charges (Figure S9).

Nevertheless, the author noted that the THz spectrum can be interpreted by either the charge formation or excitonic polarizability. In principle, the contribution from the two excitation species may be distinguished by further analysis of the THz response spectrum, i.e., the traces probed at the maximum field change or zero crossing (e.g., Kumar et al., *Sci. Adv.* 2020, 6, eaax8221). These analyses may provide more convincing results to justify the free charges.

We thank the Referee for bringing this reference to our attention. We agree that the analysis by Kumar et al., while simple, would be a means of providing a small amount of further evidence for charges, (in addition to our terahertz spectra and already taken kinetics), in that theoretically there should be a slight difference in the kinetic trace taken at the max field position compared to the zero crossing position. However, unfortunately, we are certain that this difference in kinetics would be too small for us to measure with our current setup, or likely in fact any other terahertz setups in the world currently. From our measured optical-pump-terahertz-probe spectrum in Fig S10b, we know that, at the lowest fluences we could measure, where the most charges would be present, the signal is still dominated primarily by the imaginary component, and

thus the kinetic at maximum field would also be primarily affected by a phase shift with the same kinetic as that taken at the zero crossing.

Our signal-to-noise ratio can be improved by increasing pump fluence, but we know that this physically results in a higher exciton:charge ratio, and also leads to the exciton and charge kinetics becoming more difficult to distinguish, due to the high amount of exciton and bimolecular charge recombination present at such fluences. At higher fluences there is also the risk of sample degradation and thermal effects. Even at our lowest fluences measured by Thz probe, we are in a regime wherein charges might be expected whether our hypothesis were true or not, due to rapid exciton-exciton annihilation. This is a problem common to most studies of organic materials using terahertz probes. In the terahertz literature, pump fluences are usually an order of magnitude higher than those we have reported here.

We agree that a more comprehensive study of the terahertz components may yield some more useful information on the photophysics of Y6, but such a study would still be dependent on choice of physical model for the terahertz spectra and kinetics, and we believe this study is outside the scope of our already large set of data in the present paper.

We have added a note on this point to the supplementary information, line 345:

“Further study on the dynamic terahertz properties of Y6, if able to be performed at sufficiently low signal-to-noise levels, may reveal more useful information.”

3) Saha model (Equation 1) has been adopted as the major tool to interpret the experimental data. Saha model was originally developed assuming thermodynamic equilibrium. Simply, this is not valid at all the temporal stages for ultrafast transient optical measurements. I suggest the authors to comment on this issue and clarify the scale when the model is valid.

We have not used the Saha equation as a primary tool to interpret our experimental data and apologise that this is the impression the Reviewer has taken from our manuscript. We have only compared the Saha model to our own independent model for the specific case where we have explicitly modelled the steady-state proportions of excitons and charges. We have arrived at those steady-state values by fitting dynamic data to our simple kinetic model (valid at any time, t , after photoexcitation) at multiple time points. We then ‘let the kinetic model run’ to a steady state condition, to arrive at the predicted steady state proportions of charges and excitons. We compare these proportions at different carrier densities to the Saha equation, and show that if certain reasonable parameters are assumed, i.e. effective mass and exciton binding energy, then the two approaches can yield the same results. However, our model analysis is completely independent of the Saha model.

We have clarified this point in the main text (approximately line 275), and show the relevant section below:

“The red line in Figure 3d illustrates the fraction of steady-state free charges present based on the steady-state solution to the basic kinetic model with no TTA or hole trap filling. Almost completely overlapping this line is a blue curve given by the Saha equation, with an excitonic binding energy of 270 meV (corresponding to an exciton effective mass of $\sim 0.5 m_e$). The very close match between the two curves shows that the steady-state solution of the kinetic model can be equivalent to what would be expected from the Saha equation.”

The equivalence to the Saha equation highlights the parallels of Y6 to the early days of lead halide perovskite research, where the Saha equation was also utilised as a rough approximation – once perovskites were shown to be primarily charge producing, rather than excitonic, their solar cell architecture was redirected away from primarily dye-sensitised solar cell-type designs towards planar architectures, which eventually resulted in increased efficiencies.

Reviewer #2 (Remarks to the Author):

The authors have reported intrinsic charge generation in Y6 molecule, which is one of the most popular OPV materials currently. Both spectroscopic analysis and theoretical calculation point towards the splitting of excitons in the bulk of Y6, without a molecular heterojunction. I first of all agree with other reviewers that generation of free charges in single component D-A type organic film is not new, but I do find the experimental results of this work fundamentally interesting and can potentially attract broad readership from optoelectronic community, as it offers alternative path for potential new type of optoelectronics.

The authors have improved the technical part as suggested by other reviewers, but I also agree with other reviewers the impact of this work is not properly revealed. I don't feel the data authors presented in latter part of this work actually demonstrated helpful insight to either explain currently Y6-based highly efficient OPV/to guide future optimization of OPV, nor do I feel the current results challenge mechanism of current Y6 based OPV, as written in the abstract. At this stage, I can't recommend to accept this manuscript in Nat. Commun. without major revision. Some data needs further clarifying and the way authors deliver the insight in latter part of this work needs to be re-shaped. Some questions and suggestions below perhaps to help improving the quality of the manuscript.

We are pleased that the Reviewer finds the results of our work to be fundamentally interesting to a broad readership in the optoelectronics community. The results in our paper challenge the currently understood mechanism of Y6-based bulk heterojunction OPV devices, in that we have shown that free charges are produced intrinsically in Y6, and do not need to travel to an interface of donor and acceptor to be separated. This is clearly different to the currently understood paradigm of excitons travelling to a D:A interface for charge separation, as we have already stated in the manuscript.

Based on the Reviewers suggestions, we have clarified the device section of the paper, and provided more context in the main text on a number of points, to aid readers in their understanding of the main results. We specify these changes below.

1. The author claims the mechanism of free charge generation in Y6 film is due to strong coupling between exciton and CT state, I would therefore expect a sort of hybridization of these two signatures in TA, perhaps forming a new signature. As a matter of fact, the charge signature of Y6 pristine film show the same pattern as compared to that in Y6:poly-TPD in figure 2b, the authors

need to comment on this phenomenon (is it possible to directly excite the CT state?). Additionally, can authors comment on the effect of excitation energy upon charge yield? Figure S7 shows the exciton kinetics doesn't change upon excitation energy. Once excitons are coupled with CT states to generate charges, it is hard to believe excitation energy doesn't affect this process at all (see *Polymer*, 52 (2011), 4397-4417).

The Referee raises three related points, which we now address in turn.

The author claims the mechanism of free charge generation in Y6 film is due to strong coupling between exciton and CT state, I would therefore expect a sort of hybridization of these two signatures in TA, perhaps forming a new signature.

The exciton-CT coupling in Y6 is strong in the sense that it provides access to the CT/charge states from the exciton, but not especially large in absolute terms (only 1 of the 7 pairs exhibits a coupling of >50 meV). The couplings calculated for Y6 molecular pairs are comparable to those of typical donor/acceptor systems.

Therefore, while we expect a degree of hybridisation to be present in Y6, it is unlikely that we would be able to unambiguously identify it in the TA spectra. For example, it may be that our exciton signature actually represents a hybridised exciton-CT state. However, the key result of our paper (free charge formation) does not depend on this distinction.

Hybridisation would be more clearly seen in the steady state UV-Vis absorption spectrum, although the effect would still be subtle. Our DFT calculations of molecular pairs (from which the coupling values were derived) reveal that hybridised states result in weak absorption below the main absorption band (a predicted spectrum for the pair with the largest coupling is provided below). These predictions are consistent with the broad absorption spectrum of Y6 observed experimentally, which decays slowly at low energy.

We have noted this possibility by including the sentence in the computational section, near line 380:

“Evidence of exciton/CT hybridisation may be seen in the broader red tail of the UV-vis absorption spectra.”

As a matter of fact, the charge signature of Y6 pristine film show the same pattern as compared to that in Y6:poly-TPD in figure 2b, the authors need to comment on this phenomenon (is it possible to directly excite the CT state?).

The close match between the charge signature in Y6:poly-TPD and the charge signatures we have obtained from other measurements is very much as expected, and strengthens our conclusions. We performed these TA measurements to identify what the Y6 electron signature looks like, since poly-TPD can accept a hole from Y6. The 2 spectral patterns are very similar, as would be expected because both are dominated by charges in Y6. So the close agreement between the charge signatures in the Y6:poly-TPD blend and pristine Y6 confirms our spectral assignment.

We note that there are slight differences in the two charge spectra, due to different levels of electro-absorption present, and the fact that the charge signature in neat Y6 involves holes in addition to electrons. Indeed the spectra, while very similar, are not exactly the same.

As requested, we have clarified our description of these measurements on line 171 of the manuscript:

We confirm that the orange species is that of the polaron – referred to as free charge (FC) for consistency in the rest of the manuscript – by comparing the TA spectra of Y6 blended with two hole accepting species (PTB7-Th and poly-TPD), to reveal the Y6 electron signature. The charge spectrum in neat Y6 closely resembles charge spectra obtained from these measurements (we note that the donor hole signature is also present in the PTB7-Th:Y6 measurement, which must be accounted for when making this comparison).

While it is theoretically possible to excite directly a CT-like state, as we mention above, this is not relevant to our conclusion of free charge generation, and would be extremely challenging to measure.

Additionally, can authors comment on the effect of excitation energy upon charge yield? Figure S7 shows the exciton kinetics doesn't change upon excitation energy. Once excitons are coupled with CT states to generate charges, it is hard to believe excitation energy doesn't affect this process at all (see *Polymer*, 52 (2011), 4397-4417).

We first note that Wang also presents excitation energy-dependent measurements of Y6 showing a similar independence to our own (as stated in the manuscript). This is expected from a theoretical perspective: when coupled excitons and CT states have similar energies, the hybridised states all possess the same proportion of exciton and CT character (for example Figure 3b and associated discussion in *J. Am. Chem. Soc.* 2016, 138, 11762, provided below for ease of reference). In such a situation, the ratio of photogenerated excitons and CT states are not expected to change with excitation energy. The lack of an observable dependence of charge generation on excitation energy (that we observe experimentally) is therefore consistent with the small exciton-CT energy gaps (that we calculate by TD-DFT).

Figure 3. Qualitative energy level diagrams illustrating mixing between $k = 0$ FE and CT states in the weak (a) and strong (b) coupling regimes. The FE character of the eigenstates, $|\langle k = 0 | k = 0, \alpha \rangle|^2$, is represented in red and the CTE character is represented in blue. The diabatic states (before mixing) are shown on the left, while the mixed states are shown on the right.

In light of the referee's concerns, we have modified our original statement in the paper, near line 208 to now say that the excitation energy differences are too small to be measured by our techniques, rather than to state that there is no difference at all. The relevant text now reads:

The singlet and charge species, as shown in Figure 2d, show a prompt, fluence independent (Fig. S8), interconversion within the first 2.5 picoseconds, which also has no clear excitation-energy dependence (observable within our excitation energy resolution - Fig. S8).

We note also that the paper cited by the referee (Polymer, 52 (2011)) is a ten year old review of transient absorption in polymer-fullerene-based OPV. The polymer-fullerene systems are very different to our single component non-fullerene films. They have a large degree of disorder, and low dielectric constants, unlike films of Y6, which have a very high dielectric constant, and comparatively, very low energetic disorder. With these polymer-fullerene systems, we agree that there is well documented excitation energy dependence of charge generation in blends, where the order of the blend interface has been shown to have a crucial effect (i.e. <https://www.nature.com/articles/s41467-017-02457-5>), however these considerations are no longer as relevant for single films, which helps explain the clear results, of two research groups, showing that the excitation energy dependence is very small in Y6.

2. By looking at the charge kinetics in figure 3c, the lifetime of charges seems to be relatively short (averagely around 100-200 ps). The authors claim there is trap-assisted recombination and

bimolecular recombination, and such loss mechanism happens after 100 ps (open circle in figure 3c, presumably in linear region of excitation intensity?). If this is true in BHJ (considering the neat phase), the carrier lifetime in BHJ should also be short, can authors provide similar measurements in normal PM6:Y6 BHJ? It's hard to believe highly efficient Y6-based OPV could suffer from such loss as much as shown in figure 3c. On the other hand, figure 4c shows once a molecular heterojunction is presented, recombination happens as fast as 10-20 ps. It looks like a molecular heterojunction is a "minus" for the survival of free charges here, because there is additional loss mechanism such as geminate recombination (authors need identify the recombination mechanism here)? My question is what can be gained from putting a hole acceptor here if the intrinsic generated charge is as high as 90%?

The Reviewer asks a number of related, but separate questions relating to the interpretation of exciton and charge kinetics in the latter part of the manuscript. We respond to each of these questions point by point:

"The authors claim there is trap-assisted recombination and bimolecular recombination, and such loss mechanism happens after 100 ps (open circle in figure 3c, presumably in linear region of excitation intensity?). If this is true in BHJ (considering the neat phase), the carrier lifetime in BHJ should also be short,.."

Our claim is more specific, in that there is hole-trap-assisted recombination, along with bimolecular recombination (as shown in our Jablonski diagram in Figure 3). Importantly, it does not follow that the BHJ charge lifetime should be short if the charge lifetime in the neat film is short. This is because, for both bimolecular charge recombination, and hole-trap (or n-dopant) induced recombination, if the holes are rapidly removed from the Y6 - as they are in normal BHJ blends (as shown in Fig S21, and other literature studies) - then the electrons remaining in Y6 have nothing to recombine with, which means that their recombination kinetics will be significantly slowed after electron and hole separation. This is exactly what our data shows - as can be seen by comparing the charge kinetics in Fig S20 to the neat charge kinetics. Similarly, Wang et al. (J. Am. Chem. Soc. 2020 142, 12751) demonstrate that in PM6/Y6 blends the time scale of charge formation is approximately 15 ps, i.e. much faster than the hole-trap-assisted recombination we observe. We include this point now in the blend TA section, line 328, in the paper

"For both bimolecular charge recombination, and hole-trap (or n-dopant) induced recombination, once holes are removed from the Y6 - as they are, rapidly, in normal BHJ blends (as shown in Fig S20) - then the left-over electrons have nothing with which to recombine, and their recombination kinetics is significantly slowed after electron and hole separation."

(We should also point out that the blend and device specific TA measurements, to enhance our signal-to-noise ratios, have been performed at slightly higher fluences than the lowest fluence neat film measurement.) This is also covered in slightly more detail below.

"...can authors provide similar measurements in normal PM6:Y6 BHJ?"

We have provided detailed data on PCE10:Y6 blends, in donor acceptor ratios of 1:100, 1:50, (Fig 4c) and 1:1.2 (Fig S20). These measurements show the same trends and results as what would be expected for PM6:Y6 blends (increasing charge recombination times with increasing donor content). Indeed, TA measurements of these PM6:Y6 blends in literature show that charges live

longer once they are separated (see for example, the supplementary information of Zhang et al. <https://www.nature.com/articles/s41467-020-17867-1> , and Wang et al. J. Am. Chem. Soc. 2020 142, 12751).

“It’s hard to believe highly efficient Y6-based OPV could suffer from such loss as much as shown in figure 3c.”

The Referee is correct that such highly efficient blends do not suffer such loss. As we specified above, and noted in the manuscript, the bulk heterojunction serves to reduce recombination by enabling fast spatial separation of electrons and holes into different phases.

“On the other hand, figure 4c shows once a molecular heterojunction is presented, recombination happens as fast as 10-20 ps. It looks like a molecular heterojunction is a “minus” for the survival of free charges here, because there is additional loss mechanism such as geminate recombination (authors need identify the recombination mechanism here)?”

The measurements of figure 4c were performed at twice the excitation density compared to the lowest fluence measurements of Figure 3 (5×10^{16} vs 10^{17} cm^{-3}), therefore the two measurements should not be compared directly. Our focus was on comparing the films with different amount of donor blends (i.e. the 1:50 vs the 1:100 ratio blends), therefore we had to increase the fluence slightly (compared to the lowest fluence measurement we were able to take of a neat film) to be able to resolve the different donor and acceptor kinetics. For these two measurements, and the 1:1.2 blend ratio shown in the SI, the trend is very clear, in that the more donor that is present in the blend, the longer-lived the charges are, and therefore the heterojunction is a “plus” for the survival of charges. We have thus presented a consistent account of the recombination present in both neat films and blends. The key charge recombination processes are illustrated in the diagram in figure 3: radiative and non-radiative bimolecular recombination, and recombination through (hole) traps. When a donor is added, these recombination processes are slowed, resulting in longer lived electrons in the Y6 domains.

The new clarification provided above, at line 328, is now provided in the paper to clarify our interpretation.

“My question is what can be gained from putting a hole acceptor here if the intrinsic generated charge is as high as 90%?”

Our above answer explains the current role that the hole acceptor (electron donor) plays in these blends: separating charges in different material phases slows recombination and thus enhances their lifetimes.

3. Another question related, as in response to reviewer, authors answered the second question of referee #1, saying “Additionally, while the single component device efficiency is currently low (as would be expected due to not having a p-n junction)”. My question is: the recombination mechanism presented in Y6 pristine film quenching free charges should also be presented in BHJ solar cells (geminate recombination in mixing phase and trap/bimolecular recombination in pure Y6 phase), do we even want a p-n junction in terms of keeping charges alive? The “60-90%” free

charges generated intrinsically in Y6 film don't survive much beyond 1 ns in either Y6 film or diluted BHJ, I guess a normal BHJ is still the optimal option, as authors said by themselves in page 13, the donor's key role to reduce charge recombination and split excitons. Then what's the insight to be gained from this work in terms of OPV?

We thank the reviewer for this comment. We need to clarify that we meant the donor's role is to reduce charge recombination, not to split excitons. We have modified the manuscript on line 443 that now reads:

“This hints that for current Y6-based devices the key purpose of the bulk heterojunction is more to reduce charge recombination, than to split excitons.”

This is the insight for OPV - charge formation is not limited to the donor/acceptor interface, so the donor material is not playing the role that it normally does. With this knowledge, researchers may be able to use different strategies to reduce recombination, such as chemical doping or crystal structure engineering to reduce charge recombination. Both strategies are mentioned in the paper.

As discussed above, in blend systems, recombination is not as severe because hole transfer to the donor is much faster than hole-trap-assisted recombination. Additional factors which may reduce recombination in blends are:

- 1) hole traps may be quenched by the polymer, resulting in a lower trap density
- 2) holes may be electrostatically guided to the interface by level bending (as outlined in the final section of the paper), enabling them to reach the polymer and escape trap sites present in the Y6 domains.

A p-n junction is a necessary, vital part of all high-performing solar cells, and in fact even bulk heterojunctions can be considered p-n junctions (as the donor polymer is usually p-type, and the electron acceptor is usually n-type). We mention in the manuscript that our finding opens up the possibility of making a new type of organic p-n junction, which could be more controllable than the bulk heterojunction. These junctions do indeed keep charges alive, in that an electron in an n-doped region lasts longer than it does in a p-doped region.

4. In general, Y6-based OPV is already highly efficient. The charge generation (exciton to free carrier) is not an issue anymore here. What can be further improved is the non-radiative recombination in the device. Currently, I am not sure how those highly efficient OPV can utilize the larger fraction of intrinsically generated free charge phenomenon here (plus the major trap/bimolecular recombination presented). Additionally, I think the device part is just similar compared to those Schottky junction devices. The authors have done a lot in spectroscopic part, but the device part doesn't prove any insight for the OPV community to push the envelope of device efficiency. I would urge the author to dig more in device part as to give clear information how Y6-based OPV can be further improved. The spectroscopic part alone, otherwise, should go to a more specialized journal.

This is a proof-of-concept study, demonstrating that free charge formation occurs to a significant extent in neat Y6 - an important new physical insight. Translating this knowledge into improved device performance is the subject of future work, although we have offered multiple concrete suggestions in the current paper, particularly in the final section, "Implications for Future OPV Systems".

We have also added the following clarification on Line 445: "Emphasis may shift from not only reducing interfacial and CT state bimolecular recombination, but to also focussing on a reduction in minority-carrier recombination."

The key point for device optimisation is that in normal blends, recombination is limited to the donor/acceptor interface. However, we have shown that recombination can also occur in pure Y6. So even though BHJ solar cells are able to overcome recombination in neat Y6 domains (as discussed above), the use of a blend is now not the only approach to dealing with this. We have suggested two possible strategies - which are not possible in excitonic materials - to achieve higher efficiency single component solar cells (doping and crystal structure engineering). These new options for solar cell optimisation deserve exploration.

5. For the single component Y6 device, are authors sure the charge generation mechanism is the same as pristine film without electrode? Once electrodes sandwich Y6, the difference of work function of these metals can provide electric field, which can split intra/inter-molecular CT states, as already reported in other materials.(J. Phys. Chem. Lett. 2018, 9, 8, 1885–1892) Perhaps check bias-dependent EQE? Similarly, the dilute BHJ also doesn't prove any new insight how we can utilize the results in spectroscopy, as there are also similar report showing similar effect.(J. Phys. Chem. Lett. 2020, 11, 14, 5610–5617) I urge the authors to take a re-consideration how the device part could be presented as to reveal the impact of spectroscopic part.

We are sure that the charge generation fraction is at least as high in the devices as it is in the film. We are very aware of the possibility of field-induced CT-state splitting, and it was for this reason that we thought it important to prove that 60-90% of excitons split into free charges in a neat film (without any fields present), before we presented any supporting device data. Simply put, if the charges split with zero field present, then they will also split if there is a field induced by the electrodes which aids separation

Once we are able to harvest the photogenerated charges in a single component device efficiently, it may in future be useful to attempt to quantify whether charge yields are slightly higher in a device compared to a neat film, using techniques such as bias-dependent EQE (as the Reviewer suggests) and other device-specific techniques. However, we know that any difference in charge generation yield between film and device would necessarily be small (10-40%). We have used the device section to highlight that charge recombination, rather than charge generation, is the key reason for the low efficiencies of single component devices.

The J Phys Chem Lett paper (J. Phys. Chem. Lett. 2020, 11, 14, 5610–5617) cited by the Reviewer is substantially different to our study. It is performed with a small molecule inside a fullerene matrix, which is a very different physical system to our own (a polymer in very small amounts inside a non-fullerene film). As well as the materials being different, our results are also different, in that the high absorption of the non-fullerene, and high intrinsic charge generation fraction,

likely gives rise to the unexpectedly high EQEs shown when only a very small amount of donor polymer is present. Our EQEs and device efficiencies are thus much higher than those referred to in the cited study (10% compared to ~1-2%), More study of these devices will be the subject of future works.

We thank the Reviewer for making us reconsider how the device section can be better related to reveal the impact of the spectroscopic sections. We have added a number of clarifications and motivations to the device section, to better make clear how it supports the conclusions of the spectroscopic data. We have also added some extra sentences regarding implications for devices.

We have now started the device section (line 293): “High intrinsic charge yields suggest a new path for OPV design by creating single component devices, or devices with very low donor contents (which may also be beneficial for semi-transparent OPV).”

We have added the following statement (line 303): “By replacing PEDOT:PSS with PCP-Na, whose HOMO better aligns with that of Y6, the highest PCE obtained was 0.63% (Fig S17). This value is low (though there is no junction in the active layer to separate charges and prevent minority carrier recombination).”

We have motivated more clearly the low donor content section (line 317):

“To further emphasise the importance of high charge recombination, rather than exciton splitting, as a limiting factor in device efficiency, we investigated the effect of ‘doping’ our Y6 material with very small amounts of donor polymer, PTB7-Th.⁵¹ These very dilute blends allow us to temporally separate the processes of charge separation, transport, and recombination in transient absorption measurements. We measured the photovoltaic external quantum efficiencies of the corresponding devices, which both show surprisingly high efficiencies.”

We also draw the Reviewer’s attention to the other section of the paper, which we feel is also of clear relevance to current BHJ devices, on page 16: **“Charge generation can aid high efficiency bulk heterojunction devices”**

We have also added to the section in **“Implications for Future OPV Systems”**, page 17, line 445.

“Emphasis may shift from not only reducing interfacial and CT state bimolecular recombination, but to also focussing on a reduction in minority-carrier recombination.”

Reviewer #3 (Remarks to the Author):

In the paper "Free Charge Photogeneration in a Single Component High Photovoltaic Efficiency Organic Semiconductor" the authors show that neat films of the small molecular acceptor Y6 exhibit intrinsic free charges generation. Single component devices built with Y6 have low efficiencies, mainly due to bimolecular charge recombination and intrinsic hole traps. Charge generation is argued to happen due to strong coupling between exciton and CT states and an intermolecular polarization pattern that drives exciton dissociation.

The proposed work concerns an interesting and timely research topic, lately several papers were published on single component OPV but the photophysics in many of them has not been sufficiently investigated, lacking essential information regarding charge carrier dynamics. The authors provide a very comprehensive study, delivering transient measurements, simulations as well as devices characterization. The main result stems in the simulation, from which it is argued that charges occupying distinct crystal positions, due to packing geometry, have different energetics, resulting in a bimodal density of states particularly for holes. These observations indicate that the packing geometry creates a donor-acceptor system through supramolecular, rather than synthetic means. In a way this result is similar to 10.1038/s41467-020-18439-z where it is proposed that crystalline domains with different orientations generate an electrostatic landscape with an interfacial energy offset of 0.4 eV, which promotes the formation of hybridised exciton/charge-transfer states at the interface, dissociating efficiently into free charges. Unlike this last work, though, in the submitted paper there are no different crystalline domains, rather the different energetics happens at distinct crystal positions. Still, in my opinion this point deserves at least a short discussion in the paper.

We are pleased the Referee finds the work timely, interesting, necessary and comprehensive.

We thank the Referee for pointing out the need for more comment on the differences between our work here and the published work shown in 10.1038/s41467-020-18439-z. We have added the following comment on page 15 of the paper (line 396):

“These distributions confirm the presence of a driving force for charge formation inherent to the Y6 packing structure, on the order of the exciton binding energy. This situation bears some similarity to single component OPVs in which crystalline domains with different orientations possess distinct charge energetics, resulting in charge formation at interfacial boundaries.⁶¹ The present work is distinguished from the prior literature in that the different energetics are inherent to the crystal structure, resulting in bulk - rather than interfacial - charge generation.”

Apart from this I only have a small remark: - pag. 12 it is stated "This value is low (though there is no built-in voltage in the active layer)". I suppose the contacts are ohmic or quasi-ohmic, so at short circuit there is a built in voltage \sim the gap of Y6, I don't understand the reason why there should be no built in voltage.

We agree with the reviewer that this sentence (line 302) is confusing. With ohmic contacts there would be a voltage at short circuit. We have removed this sentence and replaced it with a clearer reasoning for why we believe the PCE is low, shown below:

“This value is low (though there is no junction in the active layer to separate charges and prevent minority carrier recombination).”

REVIEWERS' COMMENTS

Reviewer #1 (Remarks to the Author):

The authors have made efforts to address my concerns. I'd recommend publishing the manuscript.

Reviewer #2 (Remarks to the Author):

The revision provided by the authors has addressed clearly the difference between the single component film and BHJ in terms of recombination kinetics. Although I think there is still room to improve in the device part, I guess that could be done in the future, as suggested by the authors.

I am satisfied with the response provided by the authors and recommend to accept this work for publication in nat.commun. I hope the fundamental insight from this work could push the OPV community forward.

Reviewer #3 (Remarks to the Author):

The authors response to my comments is satisfactory, therefore I suggest to publish the submitted paper in Nature Communications.

Responses to Reviewers' Comments

Reviewer #1 (Remarks to the Author): The authors have made efforts to address my concerns. I'd recommend publishing the manuscript.

We thank Reviewer #1 for their time and comments, which led us to improve our manuscript.

Reviewer #2 (Remarks to the Author): The revision provided by the authors has addressed clearly the difference between the single component film and BHJ in terms of recombination kinetics. Although I think there is still room to improve in the device part, I guess that could be done in the future, as suggested by the authors.

I am satisfied with the response provided by the authors and recommend to accept this work for publication in nat.commun. I hope the fundamental insight from this work could push the OPV community forward.

We thank Reviewer #2 for their time and comments. We are excited to improve the device performance in the next phase of our research.

Reviewer #3 (Remarks to the Author): The authors response to my comments is satisfactory, therefore I suggest to publish the submitted paper in Nature Communications.

We thank Reviewer #3 for their time and comments, particularly for bringing reference 58 to our attention, it is an important study for us to mention.